# Neurovascular Unit-Derived Extracellular Vesicles: From Their Physiopathological Roles to Their Clinical Applications in Acute Brain Injuries

**DOI:** 10.3390/biomedicines10092147

**Published:** 2022-09-01

**Authors:** Sandrine Reymond, Tatjana Vujić, Jean-Charles Sanchez

**Affiliations:** Department of Medicine, University of Geneva, 1206 Geneva, Switzerland

**Keywords:** extracellular vesicles, NVU, neuroinflammation, stroke, TBI, biomarker

## Abstract

Extracellular vesicles (EVs) form a heterogeneous group of membrane-enclosed structures secreted by all cell types. EVs export encapsulated materials composed of proteins, lipids, and nucleic acids, making them a key mediator in cell–cell communication. In the context of the neurovascular unit (NVU), a tightly interacting multicellular brain complex, EVs play a role in intercellular communication and in maintaining NVU functionality. In addition, NVU-derived EVs can also impact peripheral tissues by crossing the blood–brain barrier (BBB) to reach the blood stream. As such, EVs have been shown to be involved in the physiopathology of numerous neurological diseases. The presence of NVU-released EVs in the systemic circulation offers an opportunity to discover new diagnostic and prognostic markers for those diseases. This review outlines the most recent studies reporting the role of NVU-derived EVs in physiological and pathological mechanisms of the NVU, focusing on neuroinflammation and neurodegenerative diseases. Then, the clinical application of EVs-containing molecules as biomarkers in acute brain injuries, such as stroke and traumatic brain injuries (TBI), is discussed.

## 1. Introduction

Extracellular vesicles (EVs) are heterogeneous nanosized and membrane-bound particles. They have been shown to be secreted by prokaryotic and presumably, all eukaryotic cells [1]. Their lipid bilayer contains multiple transmembrane proteins and encloses a diverse cargo composed of proteins, lipids, and nucleic acids [2]. For a long time, EVs were thought to have one unique function: the removal of unnecessary molecules [3]. They are today considered an essential mechanism for intercellular communication, and their roles in multiple physiological and pathological processes have been reported [4]. The term “EVs” actually describes a large variety of vesicles. This heterogeneity comes from the differences in possible EV cargos, their physical characteristics as well as their biogenesis [5]. A few years ago, EVs were categorized according to their biogenesis in exosomes, microvesicles, and apoptotic bodies. Exosomes are derived from multivesicular endosomes, while microvesicles and apoptotic bodies are generated from the outward budding or apoptotic blebbing of the plasma membrane [4,6]. However, due to a lack of specific EV markers for each EV subtype and the difficulties in assigning a specific biogenesis pathway to an EV, the International Society of Extracellular Vesicles (ISEV) recommends classifying EVs according to their physical characteristics, such as their size, density or biochemical composition [7]. Thus, the general term “EVs” will be used throughout this review. Unfortunately, the heterogeneity and relatively low amount of EVs challenge their proper isolation and characterization. A large panel of methods have been developed to optimize EV purity and yield, but their efficacy depends on the EV-containing matrix and desired downstream application, among other factors [8].

Despite the technical difficulties, the number of scientific publications reporting the physiological and pathological roles of EVs has drastically increased in recent years [7]. Indeed, the extensive involvement of EVs in biological processes mediated by cell–cell communication as well as their secretion by all cell types enable research in multiple organisms and for a large variety of diseases. In addition, the interest of EVs as diagnostic markers and therapeutics has led to more than 500 patents being filed in the United States from 2000 to 2020 [9]. Concerning the brain, many studies have reported the involvement of EVs in physiological mechanisms, such as brain homeostasis, integrity, and protection, but also in pathological ones such as neuroinflammation and neurodegeneration processes, among others. Even though these functional roles of EVs are increasingly described, the current knowledge is still limited. An essential function of EVs is their ability to move from the brain to the blood circulation through the blood–brain barrier (BBB) and vice versa. Indeed, neuroinflammation and the early development of neurological disease is not easily investigable due to the difficulty of brain sampling [10]. Therefore, EVs constitute an interesting way to deepen the current understanding of the physiopathology of neurological diseases, as they can be isolated in biological fluids [10]. Consequently, EVs could also act as biomarkers for monitoring such diseases. In this review, we will outline the most recent studies highlighting the role of NVU-derived EVs in the physiological and pathological mechanisms of the NVU, focusing on neuroinflammation and neurodegenerative diseases. Then, we will focus on the clinical application of EVs in acute brain injuries, such as stroke and traumatic brain injuries (TBI) and the potential of EVs as prognostic and diagnostic biomarkers for such injuries.

## 2. EVs in the Central Nervous System

### 2.1. Neurovascular Unit Structure

The Neurovascular Unit (NVU) is a relatively new concept describing the relationship between brain cells with their environment and blood vessels. Knowing the NVU structure is essential to better understand brain physiopathology. The NVU is a multicellular complex composed of vascular cells such as endothelial cells, pericytes, and vascular smooth muscle cells, as well as glial cells including astrocytes, microglia and oligodendrocytes, and lastly neurons. All of these cell types are interdependent and act in concert with various physiological processes of the brain [11,12,13]. Under normal physiological conditions, NVU cells interaction results in a highly efficient barrier, whose main functions are to regulate the brain’s homeostasis, maintain its integrity, and protect it against insults (i.e., infections, trauma, and inflammation) [14]. A pivotal part of this complex cellular unit appears to be its ability to maintain brain integrity through junctional complexes, allowing them to be intimately and reciprocally linked to each other [14]. However, under certain conditions such as inflammation, traumatic brain injury or ischemic stroke, this sophisticated unit is compromised, enabling the passage of larger and hydrophilic molecules [13,15,16,17].

Among vascular cells, pericytes and brain endothelial cells are mostly involved in the maintenance of the NVU integrity [16]. As these cells are physically very close, they are essentially connected via gap junctions. The formation and maintenance of the NVU is, in part, regulated by pericytes through the secretion of inhibitory signals [18,19], whereas endothelial cells are brain regulators of the paracellular diffusion and transcellular transport of solutes and water due to the presence of highly performant junctional complexes [20]. Among these junctional complexes, the extensive presence of tight junctions, phenotypically specific to brain endothelial cells, ensures a restricted cellular permeability. This permeability limits compound movements from the systemic circulation to the brain. Therefore, they act as NVU protectors by preventing the entrance of toxic molecules from peripheral system to the brain, whereas pericytes have a macrophage-like cell role by ingesting soluble molecules present in the extracellular fluid via pinocytosis [21]. However, it is worth noticing that brain permeability is influenced by the interaction of brain cells.

Regarding glial cells, the most represented are astrocytes, whose end feet wrap around pericytes and endothelial cells. They are also key actors in regulating transcellular transport in the NVU and ensuring its maintenance by the specific presence of laminins [22]. In addition, astrocytes contribute to neuronal function as they have a common origin with neurons [23]. As for microglia, they are commonly regarded as NVU gatekeepers. Indeed, they constantly control the enclosing environment and are at the forefront of responding to any CNS injury [24]. Oligodendrocytes are the last cell type constituting the glial cell family. They are distinguished by their high lipid content, in particular myelin, an essential element for the efficient conduction of electrical impulses of axons in the CNS [25,26]. This feature gives them the unique ability to send signals to neurons through their myelin–axon interaction [25,26].

Finally, given the ability of neurons to recognize physiological variations (modification in nutrient supply and/or oxygen), they intervene by transmitting messages to neighboring neurons or astrocytes, enabling the activation of the required adjustment mechanisms [27]. Neurons are very specialized cells as they are able to transfer neurotransmitters via chemical synapses and ions through electrical synapses (gap junctions) [1]. Neuropeptides may be transported by neurons through dense core vesicles [1]. In addition to the NVU role in brain homeostasis and its integrity, the interaction of different NVU cells orchestrate, in a very careful way, the neurovascular coupling by the release of molecular mediators, recently identified as extracellular vesicles (EVs) [28,29].

### 2.2. Roles of EVs in the NVU

It has been reported that all cell types of the NVU release EVs [3,10]. An important characteristic of EVs is that they can cross the BBB to reach peripheral blood and later, other organs (Figure 1). EV transport is mostly mediated by a receptor-based mechanism and transcytosis [30,31]. By crossing this determinant barrier, EVs become an attractive tool for the diagnosis, prognosis or treatment of certain diseases. They can be a rich molecular biomarker source, a cell therapy surrogate or a drug delivery vehicle [32]. However, knowledge of the physiological role of EVs in the CNS is still limited [1,33,34]. Therefore, this section of the review examines the roles of EVs released specifically from cells of the NVU and their influence on brain function under physiological and pathological conditions (external stimuli, inflammation or brain diseases).

#### 2.2.1. Roles of NVU Cell-Derived EVs in Physiological Conditions

Although the roles of EVs have been widely reported in pathological conditions, EV enrichment has also pointed out their crucial role in physiological processes as significant vehicles for cell–cell communication.

Haqqani et al. were among the first to describe the type of proteins contained in the vesicles of human brain endothelial cells. In their study, vesicles were enriched in cell adhesion proteins, reflecting the peculiar function of this cell type [35], which is essentially to maintain the integrity of the NVU. In addition, a label-free proteomics study revealed cytoskeletal and adhesion protein enrichment in EVs released from human brain endothelial cells [36]. Due to the cross talk ability of EVs among different classes of brain cells, it is not surprising that growth factors such as the vascular endothelial growth factor (VEGF) and fibroblast growth factor (FGF), which were originally found in endothelial cells, were contained in astrocytic and neuronal EVs [37,38].

Furthermore, pericytes are also implicated in the maintenance of the NVU integrity as they work in concert with endothelial cells. In the NVU, pericytes are highly abundant and possess stem cell-like characteristics, mainly based on their repair system in response to injury [21]. As such, Yuan et al. hypothesized that EVs secreted by pericytes would reflect the natural function of pericytes, such as blood flow regulation [39] or NVU maintenance [21], to become a potential therapeutic source in the treatment of spinal cord injury (SCI) [40]. They demonstrated that pericyte-derived EVs used as possible treatment could promote blood flow, enhance endothelial function (particularly under hypoxic conditions), attenuate the apoptotic response, and thus improve recovery after SCI [40].

In essence, astrocytes have a unique role in forming the BBB, regulating brain homeostasis and supporting functions. EVs shed from young astrocytes were found to effectively sustain oligodendrocyte differentiation, whereas this support was lacking by aged astrocyte-derived EVs. This suggests that the EVs of young astrocytes have a supporting function under normal physiological conditions [41]. A comprehensive proteomic analysis in the same study revealed that only EVs released from young astrocytes contained the protein tyrosine phosphatase zeta (PTPRZ), known to participate in oligodendrocyte precursor cell (OPC) maturation [41]. The functional difference in EVs of both maturation stages suggests the ability of young astrocyte-derived EVs to transfer proteins and reinforces their support for OPC maturation [41]. Moreover, the involvement of astrocyte-derived EVs in neurite development was suggested by Wang et al. due to the presence of synapsin-I, a structural protein interconnecting synaptic vesicles to membranes [42,43]. Apart from their role in supporting and contributing to the development of brain cells, astrocytes also have a neuroprotective role. Under certain conditions such as oxidative stress, astrocytes have a neuroprotective role through their activation [44,45]. However, it was also noticed that, under normal conditions, astrocyte-derived EVs are enriched in a classical neuroprotective protein, the apolipoprotein D (ApoD) [46,47]. It is transferred to neurons and thus should enhance functional integrity and neuronal survival [48]. Other studies have consolidated these findings, as proteins and miRNAs contained in ADEVs promoted neuronal survival and neurite outgrowth [43,49].

Similar to astrocytes, EVs from microglia are also associated with neuroprotective processes. Indeed, it was proposed that the cargo of microglia-derived EVs contained important miRNAs with a neuroprotective function [50]. A proteomic study of EVs from microglia provided evidence that EVs could support neuronal growth and catabolize neuropeptides (CD13), functions that parental microglial cells also exhibit [51].

Given to the transmission function of neurons in the NVU [52], it is not surprising to consider EVs from neurons as a potential source of molecular emitters. The review by Chivet et al. described the emerging role of EVs from neurons by transporting RNA, miRNA, proteins or lipids involved in synaptic changes [53]. In addition, neuronal EVs can mediate significant neurovascular communication. It has been reported that the secretion of EVs from neurons into brain endothelial cells showed enrichment in miR-132, an evolutionarily conserved miRNA that is involved in regulating brain vascular integrity, which improved vascular integrity in this particular study [54]. However, it should be mentioned that the distinction between synaptic vesicles and EV release is still complex due to some overlapping mechanisms [55].

In conclusion, EVs seem to be able to reflect the state of their parent cells by maintaining their original function under physiological conditions and thus can be used as molecular carriers for cell–cell communication orchestrating physiological processes.

#### 2.2.2. Roles of NVU Cell-Derived EVs in Pathological Conditions

##### Neuroinflammation and EVs

Neuroinflammation is a biological process by which the innate immune system of the brain is activated after an inflammatory event such as an infection, toxin exposure, a neurodegenerative disease, aging or brain traumas [56,57,58,59]. This triggers an immediate and short activation of the innate immune system, mainly characterized by the release of inflammatory mediators such as cytokines and chemokines, and by increased BBB permeability [60]. However, a prolonged and amplified inflammatory response may have a detrimental impact due to excitotoxicity or oxidative stress, resulting in BBB breakdown [61]. Those processes can cause further damage to the surrounding tissue of the initial neurovascular injury, leading to secondary brain injuries [62,63,64]. Moreover, the pro-inflammatory microenvironment created by activated microglia and astrocytes and their release of cytokines and chemokines can increase tissue injury [65]. Red blood cells lysis and excess thrombin also produce cytotoxicity, enhancing brain damage and BBB disruption [62,63,64]. Therefore, the context, duration, and timing of primary stimuli or insults will influence the severity of neuroinflammation. This leads to the production and recruitment of different types of mediators such as chemokines, cytokines, reactive oxygen species or secondary messengers released by CNS cells to expand or counteract the inflammatory state in the brain [56,59]. As the neuroinflammation involves multiple cells from the NVU using different types of cellular communication, it is not surprising that extracellular vesicles have become a hot topic given their intercellular communication function [66].

Thus, when neuroinflammation occurs, particular inflammatory mediators may be transported by EVs, notably by microglia that are regarded as resident immune cells of the CNS, to communicate the current inflammatory state. It has been reported that EVs from microglia had upregulated expressions of miR-146a and miR-125b, involved in the regulation of the NF-κB pathway as well as in microglial activation, revealing EVs as promising modulators by promoting neuroregeneration [67]. In parallel, the study of Kumar et al. demonstrated that after a trauma, EVs released from microglia, which were initially loaded with proinflammatory molecules, were able to activate other microglia. This contributes to the ongoing neuroinflammatory reply in the injured brain and to the activation of immune responses [68]. The authors highlighted the neuroprotective function of EVs, which could potentially be considered therapeutic targets for traumatic brain injury and other neurodegenerative diseases associated with neuroinflammation. A quantitative proteomic analysis shed lights on the protein cargo of astrocyte-derived EVs (ADEVs) upon different stimuli such as trophic stimulus (adenosine triphosphate, ATP), inflammatory stimulus (IL-1β), and anti-inflammatory stimulus (IL10) [69]. Upon ATP and IL10 stimuli, ADEVs carry proteins such as RPL10 and NETO1 implicated in stimulating neurite outgrowth, dendritic branching, and enhancing neuronal survival [69]. As for ADEVs’ differentially expressed proteins in response to IL-1β, they are engaged in the regulation of the immune response (e.g., C3, PTMA, and LOX) [69]. However, ADEVs upon TNF-α and IL-1β stimuli released specific cargo that, when taken up by neurons, resulted in diminishing neuronal outgrowth, decreasing neuronal firing and promoting neuronal apoptosis [70]. In line with these results, the findings from Taylor et al. underlined that EVs shed by astrocytes under thermal stress have an increased expression of Hsc70, which may have consequences for the survival of nearby neurons [71].

Although, under multiple stimuli, EVs transport immune response elements, they are also able to propagate inflammatory mediators during diseases or disorders. Indeed, TLR-4 expression in EVs was already shown to be increased, enhancing cytokines and ROS production in EVs from microglia and astrocytes. This increase resulted in a transmission of inflammation via EVs, which provides evidence in using EVs as biomarker cargos [72,73]. Additionally, EVs from brain endothelial cells in early cerebral ischemia expressed, in turn, a change in miRNAs and surface protein profiles related to cell proliferation, cell inflammation, and angiogenesis, and as such, could possibly be considered biomarkers of endothelial cell activation and brain injury [74,75].

Moreover, it is widely recognized that coagulopathy is an important factor for secondary brain injury in trauma patients, which is related to poor outcome and may be associated with neuroinflammation and enhanced BBB permeability [76]. Indeed, there is strong evidence of a reciprocal activation between inflammation and coagulation, mainly mediated by the tissue factor pathway [77]. Regarding EVs, several studies on animals and humans have shown that platelets and cell-derived EVs could have procoagulant action [78,79,80,81], which relies on the exposure of phosphatidylserine (PS) on their surface and/or tissue factors (TF), the primary initiator of coagulation in vivo [78,82,83]. Under pathological conditions, coagulant TF-exposing microparticles can directly initiate coagulation and thrombus formation by being recruited to sites of vascular injury in vivo [83,84]. In addition, EVs, upon exposure to negatively charged phospholipids such as PS, provide a catalytic platform supporting coagulation through the facilitated formation of tenase and prothrombinase complexes [78,85]. The concomitant expression of TF and PS on EV membranes has been shown to enhance the procoagulant activity of EVs, even if the mechanism of activation of TF from its encrypted noncoagulant state to active coagulant form is not clearly established [85]. In the case of traumatic brain injury (TBI)-associated coagulopathy, EVs were described in in vivo studies to act as mediators via their procoagulant activity and through platelet activation to promote inflammation and BBB disruption [86,87,88]. The investigation of small TBI cohorts of patients reported significantly increased levels of procoagulant vesicles in the CSF and blood of severe TBI patients in comparison to control samples [79,80]. These results encourage additional investigation as EVs could serve as predictive markers for coagulopathy and therapeutic targets [87].

Taken together, studies on brain-derived EVs demonstrated that they do not simply mediate the inflammatory response. Indeed, EVs associated with proinflammatory and procoagulant molecules were reported to trigger different biological processes such as the immune response or platelet activation. However, additional investigations are required to better and more completely understand the mechanisms of EVs that are involved in brain inflammatory processes.

##### Neurodegenerative Diseases and EVs

Brain diseases can occur in multiple forms: infections (meningitis, encephalitis) [89], seizures (epilepsy) [90], trauma (concussion, traumatic brain injury (TBI)) [91], vascular conditions (stroke [92]), autoimmune conditions (vasculitis, multiple sclerosis (MS)) [93], neurodegenerative conditions (Parkinson’s disease (PD), Alzheimer’s disease (AD) [94], and tumors (glioblastoma, brain tumor) [95]. While some of these brain diseases have diagnosis predictors [95], others still face the lack of effective molecular or biological markers, for example, neurodegenerative diseases (NDs). Therefore, recent studies have reported that EV secretion and the delivery of pathogenic content would be associated with the development and progression of a variety of NDs. In this part of the review, we will focus on the involvement of EVs in NDs and some other brain diseases, and we will explore the NVU cell-derived EVs’ pathogenic and neuroprotective aspects.

Given the role of neurons in trans-synaptic exchanges, it is not surprising that EVs are regarded to be a vector for the dissemination of pathological alterations in the brain. The propagation of well-described pathological proteins contained in EVs such as tau, amyloid-β (Aβ) peptide or α-synuclein has already been depicted [53]. The research conducted by Wang et al. highlighted the release and trans-synaptic transmission of Tau protein by EVs of cultured cortical neurons [96]. EVs were able to mediate the neuron-to-neuron transportation of tau protein via direct transmission, which could contribute to the spreading of tau protein involved in Alzheimer’s disease and other tauopathies [96]. The pathogenesis of Alzheimer’s disease includes another identified hallmark, the Aβ peptide [97]. An in vitro study demonstrated that the secretion of EVs from amyloid-β protofibril-exposed cells generates neuronal dysfunction such as synapse damages, mitochondrial alterations, neuronal swelling, vacuolization, and enhanced apoptosis, potentially supporting the contribution of EVs to Aβ-induced pathology [97]. Moreover, Aβ has also been studied in the brains of HIV-1 infected individuals [98]. Indeed, an increased Aβ deposition is believed to contribute to the development of the disease [99,100]. Therefore, in the study of Ibolya et al., the results demonstrated that HIV-1 exposure increased EV release from endothelial cells with an enhanced Aβ content [101]. Furthermore, they pointed out that the EV cargo from endothelial cells was transferred to astrocytes and pericytes [101], highlighting the intercellular propagation function of EVs. Finally, Emmanouilidou et al. were the first to describe the secretion of α-synuclein in EVs from neurons. Their findings suggested that the secretion of this pathological hallmark, released by a calcium-dependent mechanism, amplifies the propagation of Parkinson’s disease [102].

Astrocyte-derived EVs have also been reported to spread or exacerbate neuropathology. A quantitative proteomics study comparing brain-derived EVs from a nontransgenic (NTg) and a transgenic amyotrophic lateral sclerosis (ALS) animal model indicated that astrocyte- and neuron-derived EVs from ALS animal models carry a misfolded and aggregated pathogenic protein, SOD1 [103]. These findings propose that EVs containing misfolded and pathogenic proteins will be transmitted into recipient cells, thus contributing to the mechanism of disease propagation [103]. Another proteomics study on brain-derived EVs including neurons, astrocytes, oligodendrocytes, and microglia, put forward that ADEVs were highly enriched in a specific hub protein, the integrin-β1 (ITGB1) associated with Alzheimer’s disease pathology [104,105]. Moreover, the correlation between the ITGB1 level and pathogenic features of AD such as Aβ42, total tau, and pSer396 tau levels was shown to be significant, supporting the leading contribution of ADEVs in AD pathogenesis. Apart from ITGB1, S100A6 is another interesting hub protein specifically enriched in ADEVs [105]. This protein is highly expressed in the brain, especially around amyloid-β plaques that can induce their degradation [106]. Thus, these combined findings highlight the role of EVs as a marker of disease progression. González-Molina et al. proposed EVs as conveyers of astrocyte deteriorations in human AD. Their results showed that EVs from AD animal models and in vitro cultures of AD post-mortem patients affected NVU cells [107]. Indeed, they observed that ADEV shedding in an AD animal model increased and that ADEVs from cell cultures were enriched in aquaporin 4 (AQ4) [107]. This could lead to NVU cell deterioration, as modified expression of AQ4 is associated with impaired waste disposal and microvascular function, particularly in the brain of AD patients [108,109]. Cell death and hyperreactivity were enhanced with ADEVs in AD animal models and the in vitro cellular model, especially in the neuron–astrocyte co-cultures [85]. Furthermore, vascular deterioration (endothelium) was established in the in vivo model induced by ADEVs from in vitro cultures [107]. Overall, these results support the detrimental effect of ADEVs in in vivo and in vitro models on the NVU components, contributing to the understanding of Alzheimer’s disease and giving novel insight into the underlying mechanisms of ADEVs in NVU degeneration.

A recent study on microglia-derived EVs from AD human brain tissue revealed disease-associated signatures. Multi-omics analysis underlined modification in microglia-derived EVs at protein, lipid, and miRNA levels, such as a higher abundance of tau protein, an increase in cholesterol lipids, and a significant increase in miRNA involved in the senescence pathway, reinforcing the hypothesis of microglia-derived EVs as vehicles of disease spreading [110]. Despite the involvement of microglia-derived EVs in the spread of pathologic molecules in AD, they were also shown to be involved in the transmission of Parkinson’s disease (PD) through α-synuclein, a classical hallmark of PD. EVs from microglia are involved in the cell-to-cell transmission of α-synuclein. To support the translational aspect of this study, microglia-derived EVs isolated from the cerebrospinal fluid of PD patients have been shown to carry aggregated α-synuclein that was able to induce nigrostrial degeneration [111].

On the other hand, there is some evidence that EVs are also able to act as scavengers of a disease and thus may be considered a therapeutic tool. Li et al. revealed the protective role of microglia-derived EVs [112]. They used a well-described cellular model of PD (cultured neurons) that was treated with MMP+, a parkinsonian toxicant, and demonstrated that microglia-derived EVs treated with non-aggregated α-synuclein appeared to exert protective properties by attenuating MPP+-mediated neurotoxicity [112]. These results indicate the possible use of EVs as a therapeutic tool in neurodegenerative diseases. Other studies highlighted the potential protective role of EVs. It was reported that EVs derived from mouse neuroblastoma cells and human cerebrospinal fluid (CSF) can sequester extracellular amyloid-β to prevent synaptic dysfunction in cultured neurons [113]. Furthermore, the administration of EVs with glycosphingolipids exerted a therapeutic effect on mouse models of AD [114], demonstrating that EVs act as a scavenger of amyloid-β and promote the clearance of extracellular amyloid-β. In addition, a summary of the multiple proteins and miRNAs contained in NVU-derived EVs is presented in Figure 2.

Overall, brain-derived EVs have a key role both as a transmitter of pathological molecules and as a potential therapeutic target source with protective properties in neurodegenerative diseases.

## 3. EVs in Acute Brain Injuries

### 3.1. Acute Brain Injuries

In this part of the review, we will focus on the clinical application of EVs in acute brain injuries and more specifically on stroke and traumatic brain injuries (TBI). Stroke and TBI are conditions affecting the brain and are among the major causes of long-term disability, affecting around 10 million people per year for TBI and resulting in 50% of stroke survivors not regaining functional independence at six months post-stroke [115,116,117]. Stroke is characterized by a sudden loss of brain function, which is, in 85% of cases, caused by an abrupt brain blood vessel blockage (ischemic stroke) or rupture (hemorrhagic stroke). On the other hand, TBI is a heterogenous condition, which is associated with several etiologies, clinical presentations, and degrees of severity ranging from severely injured unconscious patients to mildly injured patients potentially without any lesions [116,118].

Acute brain injuries leading to CNS damage are characterized by two phases, with the first one being mainly mechanical [65]. This primary damage is due to cerebral ischemia caused by vessel occlusion and to mechanical compression of the brain caused by hematoma in ischemic and hemorrhagic stroke, respectively [62,64]. For TBI, this primary injury results from the mechanical forces of the brain’s rapid acceleration or deceleration, which damage neuronal tissue by shearing, tearing, and stretching [119]. This is commonly caused by falls or traffic accidents and may lead to hemorrhage, ischemia, and/or oedema [65]. This physical insult induces damages to the neuronal tissue, resulting in cell death [65]. Following this brief first phase, the secondary phase can last for weeks or months and affects the surrounding tissue, which is sensitive to secondary damage.

In ischemic stroke, the core of the ischemic territory, composed of irreversibly damaged tissue, is surrounded by a hypoperfused zone, called the penumbra, which is salvageable if the blood flow is quickly restored, even if neuronal damages may still occur [62,120,121]. Many mechanisms are involved in the secondary damages of the brain, such as hypoxia, excitotoxicity, oxidative stress, and BBB breakdown [62,63,64]. The extent of the secondary molecular injury cascades impacts TBI severity and contributes to the reduced life expectancy of TBI patients [122], similarly to secondary injury for intracerebral hemorrhage (ICH) strokes [63]. Interestingly, injured tissues have been reported to produce EVs, which could be involved in the transition and progression of secondary injury, thus encouraging further investigation on the involvement of EVs in such brain injuries [123,124].

### 3.2. EVs’ Involvement in the Pathophysiology of Acute Brain Injuries

Numerous studies have reported increased levels of EVs in acute brain injuries [65,79,80,86,87,125,126,127,128,129,130,131,132,133,134]. Even if it is not clear if this rise is due to an increased release or impaired clearance of EVs, it could be used as an indicator of pathological processes. In fact, as endothelial-derived EVs can act as markers of endothelial activation and dysfunction, their role in the development of stroke and TBI is important to investigate [135]. For example, in acute stroke, their study could bring new insights to the understanding of the ischemic lesion development [136]. In addition, EVs from NVU’s cells have been reported to be involved in the neurogenesis, angiogenesis, and oligodendrogenesis taking place after stroke [137]. Regarding thrombotic dysregulation, platelet-derived EVs can constitute markers of the thrombotic state and could improve our understanding of coagulation disorders [135]. As an example, procoagulant EVs have been reported to be involved in TBI-associated coagulopathy and could act as biomarkers and therapeutic targets [86,87]. For these reasons, the concentration and content of EVs have been investigated as a source of biomarkers for acute brain injuries.

### 3.3. EVs as Biomarker Cargo in Acute Brain Injuries

#### 3.3.1. EVs: An Interesting Reservoir of Molecular Biomarkers

A biomarker is commonly defined as “a characteristic that is objectively measured and evaluated as an indicator of normal biological processes, pathogenic processes, or pharmacologic responses to a therapeutic intervention” [138]. While clinical or imaging measurement can constitute a biomarker, this term is usually used to describe molecules found in body fluids [139]. Molecular biomarkers classically comprise DNAs, RNAs, proteins, and metabolites. Biomarker discovery has rapidly increased in the last decades thanks to the development of high-throughput omics technologies. Biomarkers can be used for several applications such as diagnosis, prognosis, the prediction of treatment response or even to monitor the disease stage [138]. For these reasons, biomarkers have been studied in acute brain injuries.

In this aspect, EVs are promising biomarkers, as their release, as well as their cargo and surface markers, is modified according to the type and physiological or pathological state of the secreting cell [140]. Their presence in body fluids and the protection of their cargo from degradation due to the lipidic bilayer provide important advantages for EVs as biomarker source candidates [116,121]. Subsequently, EVs from the CNS could be used as markers for the diagnostics of different CNS diseases, as well as the monitoring of disease progression or to predict specific treatment responses [141].

Their ability to cross the intact BBB provides a secretion profile reflecting the processes preceding the actual BBB disruption and allowing an improvement in early patient management [140]. This is particularly interesting for acute diseases such as stroke and TBI. Indeed, early diagnosis and management in the acute phase after a TBI event may prevent secondary injuries from the complications of brain injury [142]. Additionally, the treatment window for acute ischemic stroke is within 4.5 h, and the treatment efficacy decreases with time, thus underlying the need for early biomarker discovery [143]. Consequently, this part of the review will focus on the use of EVs as biomarkers for stroke and TBI.

#### 3.3.2. EVs as Biomarker Cargo in Stroke

##### Current Biomarkers in Stroke

As mentioned previously, ischemic stroke patients should receive a reperfusion treatment within 4.5 to enhance the likelihood of a better outcome [144]. However, those therapies can only be initiated after the exclusion of hemorrhagic patients to avoid increased bleeding. The diagnosis of ischemic stroke currently relies on neuroimaging, such as computerized tomography (CT) or magnetic resonance imaging (MRI) [145]. However, acute strokes (ischemic or hemorrhagic) are detected in 80% of patients with MRI and two thirds with non-contrast CT [146]. In addition, MRI is not available in every hospital, CT scans require radiation exposure, and both are costly and cannot be performed at the prehospital level [145]. As a result, fewer than 10% of ischemic stroke patients receive thrombolysis mainly due to late admission, uncertainty about the stroke type or potential salvageable brain tissue [147,148].

Blood biomarkers distinguishing the different stroke types or predicting lesion volume and outcome could enhance the number of ischemic stroke patients receiving reperfusion therapies [149]. For this reason, blood biomarkers have been extensively investigated in the last two decades [150]. D-Dimer, GFAP, H-FABP, NTproBNP or S100B among others appear as promising candidates for the differentiation of acute stroke types [151,152,153,154,155,156,157,158,159,160,161,162,163,164,165,166,167,168,169,170]. Other investigations focused on biomarkers predicting hemorrhagic transformation [171,172,173,174,175,176,177,178,179] or the outcome of patients [180,181,182,183,184,185,186] or even biomarkers associated with the penumbra-infarct volume [187,188,189]. Single or panels of stroke marker candidates were reviewed by Dagonnier et al. [148]. However, none is currently implemented in clinics as adequate performance has not been reached [148].

##### Potential EV-Associated Biomarkers in Stroke

In the hope of finding novel biomarkers, EVs have been investigated. Indeed, acute EV profiles could allow us to distinguish the different stroke types to have a better understanding of the cerebrovascular disease state and stroke etiology while the BBB is still intact [140]. The measurements of EVs as acute biomarkers for stroke diagnosis could even be done at the pre-hospital stage [140]. As a result, the diagnostic and prognostic performances of circulating EV levels have been investigated in stroke. As EVs display a specific molecular profile depending on their cell origin, most studies have focused on cell type-specific EV populations (Table 1).

Regarding the identification of subarachnoid hemorrhage (SAH), Lackner et al. observed that endothelial-, leukocyte-, and erythrocyte-EVs were elevated in SAH patients in comparison to healthy individuals [128]. Similar findings were observed by Sanborn et al.: SAH patients presented an elevation in endothelial cell-, platelet, erythrocyte-, and neutrophil-derived EVs levels, with a variation in the temporal profile depending on the EV subtype [129]. As for intracerebral hemorrhage (ICH), Huang et al. observed increased levels of procoagulant PS-exposed EVs in the CSF and blood of patients with ICH in comparison to controls [130]. Another study from the same team showed a similar increase in the blood of a larger number of ICH patients [191]. They suggested that plasma EV levels at admission could be a predictor of 1-week mortality in ICH patients [130,191]. Regarding stroke etiology, Kuriyama et al. observed increased levels of platelet-derived EVs in patients with large-artery atherosclerosis and small-vessel occlusion in comparison to healthy controls [131].

To accelerate its diagnosis, numerous studies focused on the identification of IS patients while investigating EV populations from different cell origins. Increased levels of endothelial-derived EVs were observed in acute ischemic stroke patients in comparison to controls [132,133,136], as well as in transient ischemic attack patients [132]. Simak et al. reported that this increase in acute ischemic stroke patients was also associated with stroke severity [136]. Li and Qin observed that while endothelial-derived EVs increased, platelet-derived EVs levels remained unchanged [133]. On the contrary, Chen et al. observed elevated levels of platelet-derived EVs in acute ischemic stroke patients. In their study, the level of platelet-derived EVs was an independent risk factor for the infarct volume of acute ischemic stroke [190]. In addition, Chiva-Blanch et al. observed increased numbers of neuronal precursor cell-, platelet-, endothelial-, and circulating immune cell-derived EVs in ischemic stroke patients in comparison to age-matched high-risk cardiovascular controls [134]. The discrepancy in the results may come from the differences in methodology (e.g., CD marker to select the EV population) and patient populations. More details can be found in the review written by El-Gamal et al. [135]. Consequently, there is a need for more clinical studies with larger sample sizes to validate those findings.

Regarding miRNA, several candidates have been discovered for ischemic stroke diagnosis (Table 2). MirR-9, miR-124, mirR-134, and miR-223 increased in acute ischemic stroke patients in comparison to controls and were correlated with NIHSS scores [121,192,193,194]. In addition, miR-21-5p and miRNA-30a-5p could differentiate the hyper acute, acute, sub-acute, and recovery phases of ischemic stroke [195]. Concerning the distinction of ischemic from hemorrhagic stroke, miR-27b-3p was higher in acute ischemic stroke patients than in hemorrhagic stroke patients, and miR-146b-5p was higher in acute ischemic stroke patients than controls and SAH [196]. In the same study, Kalani et al. identified a subset of ex-miRNAs, which could differentiate SAH from other stroke types with an AUC of 0.927 and an accuracy of 0.972 [196]. However, the use of nucleic acids such as miRNAs to be measured in point-of-care (POC) is challenged by the need to be detected in small volumes without time-intensive sequencing platforms [196]. More technological developments are required to make POC detection feasible.

For this reason, protein biomarkers are currently investigated as they could be measured in POC, for example, in pre-hospital settings, and give rapid indications allowing them to be used as decision-making tools. To the best of our knowledge, protein biomarker candidates in EVs for stroke are scarce. Some studies identified pro-inflammatory cargoes in EVs released by stroke patients (Table 2). Proteomics analysis conducted by Couch et al. revealed significant upregulation of several pro-inflammatory proteins, such as C-reactive protein in EVs from stroke patients in comparison to age-matched controls [197]. Another study investigated the protein profile of serum-derived EVs in patients who developed symptomatic ischemic stroke. In comparison to age- and sex-matched controls who did not develop any brain lesion, alpha-2-macroglobulin (A2MG), complement C1q subcomponent subunit B (C1QB), complement C1r subcomponent (C1R), and histidine-rich glycoprotein (HRG) were significantly upregulated in the ischemic stroke patients. A2MG and HRG upregulation have been associated with fibrinolytic cascade dysregulation, and all four proteins are involved in inflammatory processes and could be predictors of future ischemic stroke [198]. Finally, Datta et al. found the upregulation of proteins involved in coagulation in patients with lacunar infarction with adverse outcome [199].

To conclude, potential biomarker candidates for stroke are emerging, but additional studies in larger cohorts are required to verify them. With the aim of improving the diagnostic performance, combinations of several biomarkers should be further explored, for example, through microfluidic chips for EV miRNAs [140].

#### 3.3.3. EVs as Biomarker Cargo in TBI

##### Current Biomarkers in TBI

At admission, the diagnosis of TBI mainly relies on neurological examination to watch for diverse symptoms and on computerized tomography (CT) scan to identify brain lesions [118]. However, studies have revealed that only 10% of mild TBI (mTBI) patients, which accounts for 70–90% of all TBI events, will be CT-positive [119,142]. In addition, CT scanning presents possible adverse effects caused by radiation exposure and it is costly [200]. Therefore, better surrogate markers of brain injury are needed, and blood biomarkers are one solution that is cheaper and radiation-free [118].

Consequently, several studies have investigated biomarkers candidates, such as tau, p-tau, neurofilament light chain (NfL), glial fibrillary acidic protein (GFAP), and ubiquitin carboxyl-terminal hydrolase isozyme L1 (UCHL1) [116]. They have been associated with TBI severity and poor recovery in civilians, but also athletes and military personnel [200,201,202,203,204]. Additionally, IL-10, H-FABP, GFAP, and S100B, the most studied one, have been explored as biomarkers to predict brain lesions due to trauma [118,119,205]. In fact, a CE certificate was granted in 2019 to TBICheckTM, a portable diagnostic test measuring H-FABP to detect mTBI [206]. In 2021, another rapid mobile TBI blood test including measurements of UCH-L1 and GFAP in plasma on Abbott’s handheld i-STAT™ Alinity™ platform, received FDA 510(k) clearance [207]. S100B is used in clinics in some countries in Europe for mTBI patients with low risks for traumatic injuries [208]. However, it is not recommended for patients with extracranial injuries and for those whose trauma occurred more than 6 h previously due to S100B’s unspecificity and short half-life [208]. Subsequently, the investigation of novel biomarkers is relevant.

##### Potential EV-Associated Biomarkers in TBI

The main research on EVs in TBI patients originally focused on the understanding of their involvement in TBI pathophysiology [79,80,86,87]. In the last five years, more research was done on biomarker discovery in EVs for the diagnosis of TBI. In this section, we will provide a broad overview of the studies relating to EV biomarkers in TBI patients (Table 3). For a more detailed description of clinical and in vivo studies, the reader is referred to the review published in 2020 by Guedes et al. [116].

In most studies, significantly increased levels of circulating EVs have been observed in acute TBI patients as well as in rodents [65,79,80,86,87,125,126,127]. As for specific biomarker, GFAP, UCHL-1, and NFL were also investigated in plasma or serum-derived EVs of TBI patients due to their potential as TBI biomarkers.

Regarding GFAP, its levels in plasma-derived EVs increased in acute mTBI patients in comparison to controls, along with IL-6 [210]. A similar increase in GFAP was identified in serum-derived EVs from moderate and severe TBI patients up to five days [215] or one year after the TBI event [211]. In the latter study, the EV GFAP levels could distinguish controls from moderate and severe TBI with an AUC of 0.86 and 0.88, respectively, and were correlated with worse 1-year clinical outcomes [211]. EV NFL levels were also elevated and correlated with worse clinical outcome one year after TBI [211]. The authors suggest a possible diagnostic and prognostic utility of the EV GFAP concentration even if sampled one year after the TBI event.

UCH-L1 concentrations were elevated in moderate to severe TBI patients one day after the TBI event and decreased over time (up to five days), along with EV total-tau [215]. Tau is an important biomarker candidate, as its abnormally phosphorylated forms have been associated with several neurodegenerative diseases [214]. In fact, plasma-derived EV tau was elevated in symptomatic former football players who suffered repetitive mTBIs and met diagnostic criteria for traumatic encephalopathy syndrome in comparison to the control group, and this higher EV tau was significantly correlated with worse cognitive functioning [216]. A similar increase in EV tau and p-tau was identified in war veterans with a history of repetitive mTBIs compared to those with no or one to two mTBI events [214] or to controls [212]. Interestingly, this increase was only a trend in measured circulating tau, suggesting that EV tau levels could perform better as biomarkers than their circulating ones [214]. Additionally, Beard et al. showed that circulating brain-derived EVs and plasma exhibited distinct biomarker distributions for a panel of neurodegeneration and inflammation markers in mTBI, regrouping GFAP, UCHL1, NFL, tau, IL-10, IL-6, and TNF-alpha [210]. Winston et al. reported that plasma neuronally-derived EV (NDE) cargo proteins from mTBI soldiers induced cytotoxicity in neuron-like recipient cells in vitro, but not astrocytic-derived cargo proteins, supporting the presence of markers of neurodegeneration in NDEs of mTBI [217]. The protein cargo of NDEs was further investigated in acute and chronic mTBI athletes by Goetzl et al. [213]. The NDE levels of several functional brain proteins were significantly dysregulated in acute mTBI, such as UCHL1, aquaporin-4, NKCC-1, and synaptogyrin 3, while neuropathological proteins were increased in chronic mTBI (Aβ42, P-T181-tau, P-S396-tau, IL-16, and PRPc). These findings suggest that the cargo of plasma NDEs could distinguish between acute and chronic sports-related mTBI [116]. Elevated EV levels of Aβ42 were also found in mTBI military service members [217] and chronic mTBI veterans in comparison to controls [212]. In addition to protein biomarkers, several miRNAs relating to neuronal function, vascular remodeling, and BBB integrity were elevated in plasma-derived EVs of chronic mTBI veterans in comparison to controls [218].

In conclusion, several EV-containing proteins appear to be potential biomarker for TBI diagnosis and prognosis. However, the lack of reproducibility between the studies challenges the validation of candidates [116]. Verification in larger cohorts of patients is essential, as most mentioned studies were done in small cohorts and with different study designs.

## 4. Conclusions

In conclusion, EVs are involved in many physiological processes as well as pathological ones. In the NVU, the physiological roles of EVs are more similar than the original function of their parent cells. Thus, NVU-derived EVs seem to serve as molecular carriers for cell–cell communication under physiological conditions. Regarding their involvement in the physiopathology of neurodegenerative diseases, brain-derived EVs play a dual role. They are reported to promote disease propagation, such as AD, PD, and ALS, via the transmission of pathological molecules (Aβ, α-synuclein, tau, and SOD1). On the contrary, they could also represent a potential therapeutic target source for neuroprotection. As a matter of fact, EVs are biocompatible sources exerting great potential for therapeutics [219]. Indeed, their endogenous origin confers them enhanced advantages over synthetic nanoparticles developed in the drug delivery system. However, to fully exploit their potential, their spreading mechanisms still need to be better understood to enhance their direction to a specific target.

Interestingly, three important proteins involved in Alzheimer’s disease were identified in EVs in in vitro models and in TBI patients: tau, Aβ, and AQ4. In in vitro AD models, tau was present in neuron- and microglia-derived EVs, suggesting its involvement in AD propagation [96,110]. As for acute and chronic TBI patients, EV tau (total tau or in a phosphorylated form) was present at a higher level [212,213,214,215,216]. Regarding the Aβ peptide and its long form, Aβ42, EVs shed by Aβ-exposed cells were reported to enhance neuronal dysfunction [97], while another study suggested that neuroblastoma-derived EVs could scavenge Aβ [114]. Interestingly, astrocyte- and neuron-derived EVs contained Aβ42 at higher levels in patients suffering from repetitive or chronic mTBI than controls [212,213,217]. In addition, AQ4 was increased in astrocyte-derived EVs in an AD animal model [85] and neuron-derived EVs from acute mTBI patients [213]. Thus, TBI is likely involved in AD, as well as in PD and epilepsy among many others, and could even be a cause of AD development [220]. The relationship between AD development and TBI is very complex and still poorly understood [221]. Albeit, a deeper knowledge of the role that EVs play in AD and TBI could allow a better understanding of the development of AD after a TBI event. Additionally, EVs could possibly act as a biomarker for the prognosis and diagnosis of AD.

Moreover, EVs are also associated with acute brain injuries such as stroke and TBI, for which they could act as biomarkers for diagnosis and prognosis. Several studies have proposed potential biomarker candidates, but further research is necessary in larger cohorts to validate them. Even if EVs represent promising biomarker sources with advantages in comparison to circulating molecules, their investigation also comes with technical challenges. The lack of standards for EV isolation and characterization results in a large variety of methods, enhancing the divergence of results [116]. Moreover, differences in storage conditions, collection, and handling procedures have an impact on EV yield, purity, and integrity [222]. As a result, the reproducibility and validation of potential EV biomarker are arduous. As those difficulties are common for all applications related to EVs, guidelines and task forces were put into place to palliate to such issues, as the Minimal Information for Studies of EVs (MISEV), but more work is needed [7,222,223]. Moreover, the low recovery efficiency of classical EV isolation techniques limits the identification and quantification of low abundant EV molecules, thus restricting biomarker discovery [224]. With this aim in mind, nanotechnologies and biosensing platforms are being developed [224].

An additional concern is the implementation of EVs in a clinical setting, as those traditional methods require time and equipment and are not always fit to process a high number of samples and small volumes [140]. For this purpose, new technologies are emerging such as nano-scale resolution flow cytometry or chip array platforms, which could be used for in hospitals [222]. Regarding pre-hospital POC diagnostic tests, blood-derived EV biomarkers have significant potential, but novel POC development will be required with an EV marker measurement performed in a small volume of whole-blood, quickly and with minimal handling [140].

Overall, EVs represent a promising field of research with multiple applications. We hope that with the growing interest of the scientific community and biopharmaceutical industry, solutions to the current technical challenges will be developed in the next years.

## Figures and Tables

**Figure 1 biomedicines-10-02147-f001:**
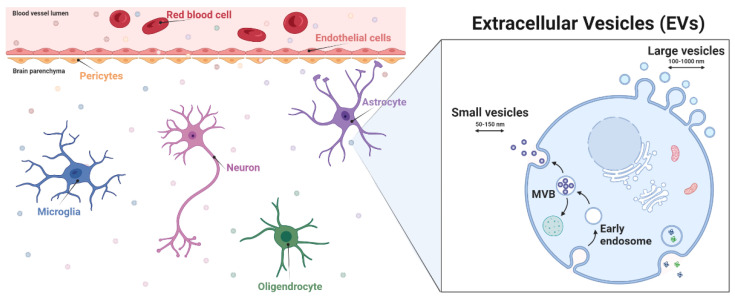
Extracellular vesicles in the neurovascular unit environment. Neurovascular unit components are represented such as brain endothelial cells, pericytes, astrocytes, microglia, oligodendrocytes, and neurons. Extracellular vesicle biogenesis representation. Figure created with BioRender.com (accessed on the 26 August 2022).

**Figure 2 biomedicines-10-02147-f002:**
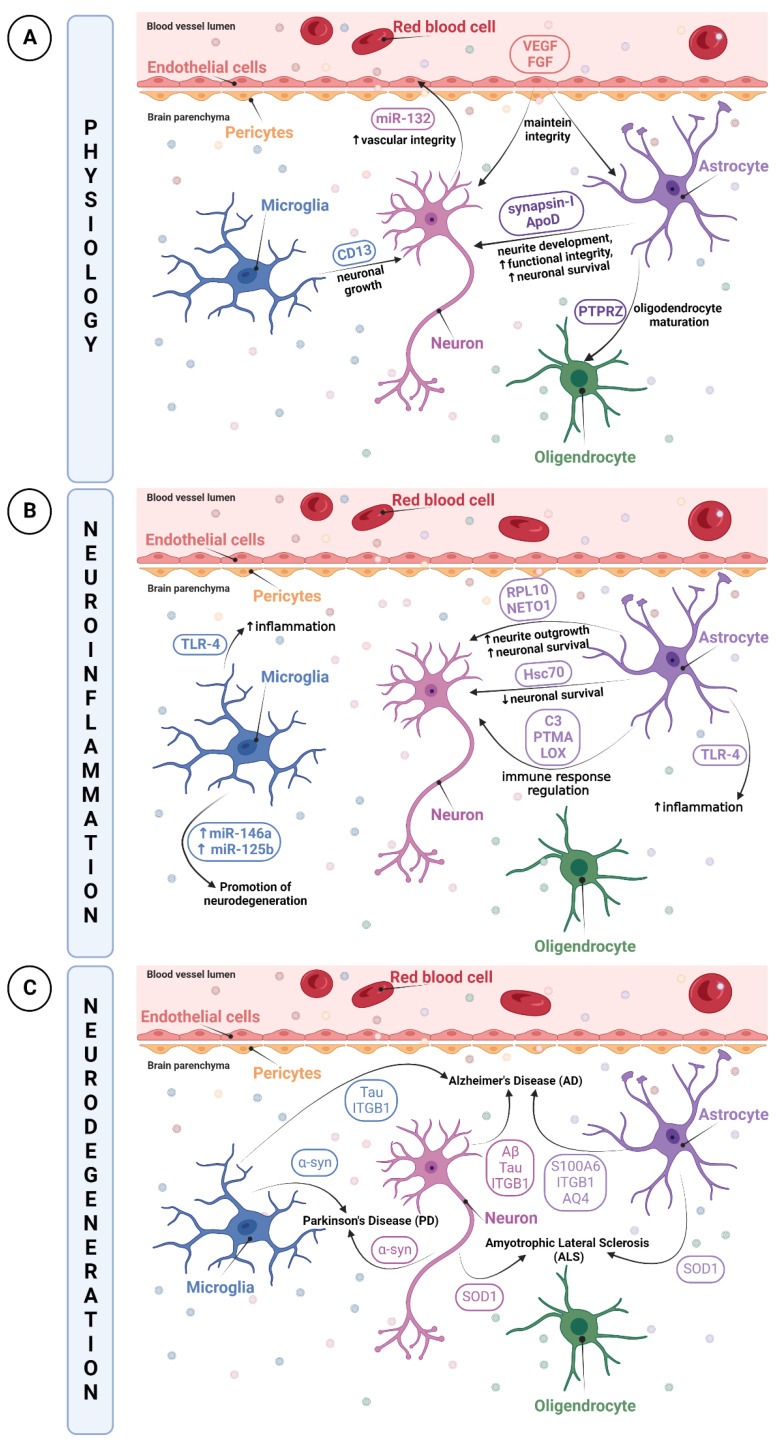
NVU-derived EV molecular content. Proteins and miRNAs contained in NVU-derived EVs are represented according to their role in (**A**) physiological conditions, (**B**) neuroinflammation, and (**C**) neurodegeneration. Proteins and miRNAs encircled in orange for endothelial cell-derived EVs, in blue for microglia-derived EVs, in pink for neuron-derived EVs, and in purple for astrocyte-derived EVs. Figure created with BioRender.com (accessed on the 26 August 2022).

**Table 1 biomedicines-10-02147-t001:** EV population as biomarker candidates in stroke.

EV populations as Biomarkers	Application	Cohort	EV Detection Method	Reference
Plasma endothelial-, activated platelet-, erythrocytes-, granulocytes, and leukocytes-derived EVs	Detection of AIS and TIA	AIS (*n* = 66) and TIA (*n* = 21) patients and healthy participants (*n* = 24)	Flow Cytometry	[132]
Plasma platelet-derived EVs	Detection of AIS	AIS patients with LAA (*n* = 53) or SAO (*n* = 59) and healthy participants (*n* = 35)	Flow Cytometry	[190]
Plasma endothelial-, platelet-, neuronal precursor cell-, circulating immune cells-derived EVs	Detection of IS	AIS patients (*n* = 44) and high-risk cardiovascular participants (*n* = 44)	Flow Cytometry	[134]
Plasma, and CSF PS+ EVs	Detection of ICH	ICH patients (*n* = 36) and controls (*n* = 10)	Biotinylation of Annexin V	[130]
Plasma platelet-derived EVs	Detection of SAO and LAA	SAO (*n* = 34), LAA (*n* = 41), cardioembolism (*n* = 20), and undetermined etiology (*n* = 15) patients and healthy participants (*n* = 61)	Immunoassay	[131]
Plasma endothelial-, leukocyte- and erythrocyte-derived EVs	Detection of SAH	SAH patients (*n* = 20) and healthy participants (*n* = 20)	Flow Cytometry	[128]
Plasma endothelial-derived EVs	Detection of AIS	AIS patients (*n* = 68) and healthy participants (*n* = 61)	Flow Cytometry	[133]
Plasma endothelial-, erythrocyte-, neutrophil- and platelet-derived EVs	Detection of SAH	SAH patients (*n* = 22) and healthy participants (*n* = 13)	Flow Cytometry	[129]
Plasma PS+ endothelial-derived EVs	Detection of AIS	AIS patients (*n* = 41) and healthy participants (*n* = 23)	Flow Cytometry	[136]

Abbreviations: AIS = acute ischemic stroke, IS = ischemic stroke, IPH = intraparenchymal hemorrhage, LAA = large artery atherosclerosis, SAH = subarachnoid hemorrhage, SAO = small artery occlusion. Inclusion and exclusion criteria are described in Appendix A.

**Table 2 biomedicines-10-02147-t002:** EV cargo as biomarker candidates in stroke.

Biomarker	Sample Type	Application	Cohort	EV Isolation Method	Reference
miR-9, miR-124	Serum	Detection of AIS	AIS patients (*n* = 65) and healthy participants (*n* = 66)	ExoQuick (System Biosciences)	[192]
miR-134	Serum	Detection of AIS	AIS patients (*n* = 50) and healthy participants (*n* = 50)	ExoQuick (System Biosciences)	[193]
miR-223	Serum	Detection of AIS	AIS patients (*n* = 50)	ExoQuick (System Biosciences)	[194]
miR-21-5p, miR-30a-5p	Plasma	Detection of IS	hyperacute (*n* = 15), acute (*n* = 55), subacute (*n* = 31) and recovery phase (*n* = 32) IS patients and healthy participants (*n* = 24)	ExoRNeasy (QIAGEN)	[195]
miR-27b-3p, miR-146b-5p	Plasma	Detection of AIS	IS (*n* = 21), IPH (*n* = 19) and SAH (*n* = 17) patients	ExoRNeasy (QIAGEN)	[196]
A2MG, C1Q, C1R, HRG	Serum	Detection of AIS	AIS patients (*n* = 38) and healthy participants	Ultracentrifugation	[197]

Abbreviations: AIS = acute ischemic stroke, IS = ischemic stroke, IPH = intraparenchymal hemorrhage, LAA = large artery atherosclerosis, SAH = subarachnoid hemorrhage. Inclusion and exclusion criteria are described in Appendix A.

**Table 3 biomedicines-10-02147-t003:** EV biomarker candidates in TBI.

Biomarker Candidates	Application	EV Population	Cohort	EV Isolation Methods	Reference
GFAP, IL-16	Detection of acute mTBI	Plasma GluR2+ brain-derived EVs	mTBI patients (*n* = 47), healthy (*n* = 39) and orthopedically injured (*n* = 7) participants	TENPO nanofluidic platform [209]	[210]
GFAP, NFL	Detection of 1-year TBI	Serum EVs	TBI patients (*n* = 72) and healthy participants (*n* = 20)	ExoQuick (System Biosciences)	[211]
Aβ42, IL-10, tau	Detection of mTBI	Plasma L1CAM+ neuron-derived EVs	mTBI military personnel (*n* = 42) vs. healthy participants (*n* = 22)	ExoQuick (System Biosciences) + L1CAM immunoprecipitation	[212]
Aβ42, AQ4, IL-16, NKCC1, P-T181, P-S396-tau, PRPc, UCHL-1, Synaptogyrin-3	Detection of acute and chronic mTBI	Plasma L1CAM+ neuron-derived EVs	Acute (*n* = 18) and chronic (*n* = 14) mTBI and healthy participants (*n* = 21)	ExoQuick (System Biosciences) + L1CAM immunoprecipitation	[213]
Tau, p-tau	Detection of repetitive mTBI	Plasma EVs	Repetitive mTBI (*n* = 56), with 1–2 mTBI (*n* = 94) and without TBI (*n* = 45) participants	ExoQuick (System Biosciences)	[214]
GFAP, tau, UCHL-1	Detection of moderate–severe TBI	Serum EVs	Moderate-severe TBI patients (*n* = 21)	ExoQuick (System Biosciences)	[215]
Tau	Detection of chronic traumatic encephalopathy (CTE)	Plasma EVs	Former NFL players with CTE and repetitive TBI (*n* = 78) and participants with a reported history of TBI (*n* = 15)	Size Exclusion Chromatography (Agarose Bead Technologies)	[216]
Aβ42, neurogranin	Detection of mTBI	Plasma L1CAM+ neuron- and GLAST+ astrocyte-derived EVs	With mTBI (*n* = 19) and without mTBI (*n* = 20) military personnel	ExoQuick (System Biosciences) + L1CAM or GLAST immunocapture + FACS selection	[217]

Abbreviations: CTE = Traumatic Chronic Encephalopathy, NFL = National Football League. Inclusion and exclusion criteria are described in Appendix A.

## Data Availability

Not applicable.

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
