# Peer review of "Neurovascular Unit-Derived Extracellular Vesicles: From Their Physiopathological Roles to Their Clinical Applications in Acute Brain Injuries"

_biomedicines, 2022, doi:10.3390/biomedicines10092147_

Round 1
Reviewer 1 Report
The article is interesting, being an excellent review on the EVs released by the Neurovascular Unit, always taking into consideration the main physiological and pathological mechanisms that these EVs can trigger. However, there are some questions that need to be clarified:
1. The abstract and the introduction are good, having demonstrated the main objectives of this work.
2. In the line 78, there are more recent reviews on the neurovascular unit, namely (Curr Neurovasc Res. 2019;16(5):502-515. doi:10.2174/1567202616666191026122642. and Brain Res Bull. 2022 Jun 15;184:34-45. doi: 10.1016/j.brainresbull.2022.03.011. )
3. In the figure 1, the smooth muscle cells are not represented, I think they are a very important constituent of the NVU and have been overlooked.
4. In the item 2.2.1. Roles of NVU cells-derived EVs in physiological conditions there is no reference to vascular smooth muscle cells, previous reviews may help.
5. Line 285 –(stroke, cerebrovascular accident (CVA)) explains the difference between these two pathologies?
6. In the figure 2, the smooth muscle cells are not represented, please include.
7. The tables are good
8. The conclusion could be improved, notably by discussing the importance of SMCs.
Author Response
Response to Reviewer 1 Comments
Point 1. The abstract and the introduction are good, having demonstrated the main objectives of this work.
Response 1: We thank the reviewer for this remark.
Point 2. In the line 78, there are more recent reviews on the neurovascular unit, namely (Curr Neurovasc Res. 2019;16(5):502-515. doi:10.2174/1567202616666191026122642. and Brain Res Bull. 2022 Jun 15;184:34-45. doi: 10.1016/j.brainresbull.2022.03.011.)
Response 2: We thank the reviewer for this pertinent comment. We included this suggested review: doi: 10.1016/j.brainresbull.2022.03.011, as the other one is not an open access one.
Point 3. In the figure 1, the smooth muscle cells are not represented, I think they are a very important constituent of the NVU and have been overlooked.
Response 3: We thank the reviewer for pointing out this comment. It is perfectly true that smooth muscle cells (SMCs) are important constituent of the NVU at artery and arteriole level but not at capillary level, as SMCs are usually replaced by pericytes. In our figure 1, we choose to focus on pericytes rather than SMCs as there is currently a lack of information about extracellular vesicles (EVs) released by SMCs in the brain, their role and their function. In our review, we aimed at targeting the current knowledge on the role of EVs released from brain cell types (from the neurovascular unit (NVU) more specifically). However, as the field of EVs is at its infancy, there is very few available data or even none information about EVs released from SMCs in the NVU. This is the reason explaining why we do not include SMCs in our figures. However, we are convinced that with the current popularity that EVs field is experimenting and with the technological progress, the information regarding SMCs-derived EVs will sooner or later be beneficial to the scientific community.
Point 4. In the item 2.2.1. Roles of NVU cells-derived EVs in physiological conditions there is no reference to vascular smooth muscle cells, previous reviews may help.
Response 4: We thank the reviewer for this remark. As mentioned in point 3 and after having extensively examined the current literature regarding EVs released from SMCs in the NVU, we were not able to add knowledge on them due to the lack of references in the brain era. We found some papers describing the influence of brain-derived EVs on SMCs but nothing directly related to the role of EVs from SMCs (https://doi.org/10.1089/neu.2021.0274). Furthermore, to our knowledge, there is only one study using EVs from brain cells such as brain endothelial cells, astrocytes, marrow stromal cells and SMCs. As their results showed that miR-126 expression was only significantly higher in brain endothelial cells and in their derived EVs, this study was unfortunately not retained to justify the SMCs-derived EVs role in our review (https://doi.org/10.1161/STROKEAHA.119.025371).
Point 5. Line 285 (stroke, cerebrovascular accident (CVA)) explains the difference between these two pathologies?
Response 5: We thank the reviewer for this comment, the duplication was corrected (line 290-291 in red). Please see the attachment.
Point 6. In the figure 2, the smooth muscle cells are not represented, please include.
Response 6: We thank reviewer for this remark. As mentioned in point 3 and 4, the current lack of knowledge on the EVs released from MSCs in the brain do not allow us to modify the figure as suggested.
Point 7. The tables are good.
Response 7: We thank the reviewer for this comment.
Point 8. The conclusion could be improved, notably by discussing the importance of SMCs.
Response 8: We thank the reviewer for this comment and we kindly invite the reviewer to understand the reasons why we have not modified the conclusion with reference to the explanations provided in points 3, 4 and 6. However, we hope that in the coming years we will have the opportunity to include the results of our colleagues on EVs released from SMCs in the brain to complement the current knowledge.

Reviewer 2 Report
Thank you for the opportunity to review this manuscript, dealing with important explanations entitled “Neurovascular unit-derived extracellular vesicles: from their physiopathological roles to their clinical applications in acute brain injuries”. Authors are reporting the role of NVU-derived EVs in physiological and pathological mechanisms of the NVU, focusing on neuroinflammation and neurodegenerative diseases, their clinical application in acute brain injuries, as well as the potential of EVs as prognostic and diagnostic biomarkers. Overall explanations strategy is good, however, there’s a lack of mechanistic overview of Figures which does not show proper information as in text material. The authors need to revise and complete both figures with mechanistic information. Though Figure 2 information is concise, it is not understandable for the general readers. Furthermore, their explanation and representation of clinical studies in tables are ok, however, there’s a lack of selection of clinical studies with inclusion/exclusion criteria. Authors need to include a flow chart representing the selection criteria in addition to the total number of samples included in these explanations with the category.

Author Response
Response to Reviewer 2 Comments
Please see the attachment for our answers.
Point 1. Overall explanations strategy is good, however, there’s a lack of mechanistic overview of Figures which does not show proper information as in text material. The authors need to revise and complete both figures with mechanistic information. Though Figure 2 information is concise, it is not understandable for the general readers.
Response 1: We thank the reviewer for these pertinent remarks. Regarding:
- Figure 1: The aim of the figure 1 was to help the readers to visualize the different brain cell types that constitute the neurovascular unit and specifically the dynamic interaction of extracellular vesicles (EVs) within the NVU. In addition, we aimed at pointing out the EVs ability to cross the blood brain barrier. Therefore, we have slightly modified this figure to be more consistent. Indeed, the EVs are no longer in blue but in the same color as their parent cells.
- Figure 2: We have extensively revised figure 2 based on the relevant suggestions of the reviewer and we hope that this proposal will better reflect the information in the text.
Point 2. Furthermore, their explanation and representation of clinical studies in tables are ok, however, there’s a lack of selection of clinical studies with inclusion/exclusion criteria. Authors need to include a flow chart representing the selection criteria in addition to the total number of samples included in these explanations with the category.
Response 2:
We thank the reviewer for this comment. For each clinical study cited in the tables, inclusion and exclusion criteria were added in Supplemental Tables 1, 2 and 3 (at the end of the attachment). Regarding the request for a flowchart, we would like to point out that this review is not a systematic one and is not based on an approach representable in a flowchart. Moreover, the collection of studies reported in this review is not exhaustive and we explicitly recommended other reviews to the readers to complement ours (lines 550-552, 621-622, both in red in the text).
